# Countries' progress towards Global Health Security (GHS) increased health systems resilience during the Coronavirus Disease-19 (COVID-19) pandemic: A difference-in-difference study of 191 countries

**Tyler Y. Headley**[1], **Sooyoung Kim**[2], **Yesim Tozan**[3]*

**1** Department of Political Science, New York University Abu Dhabi, Abu Dhabi, United Arab Emirates, **2** Department of Public Health Policy and Management, School of Global Public Health, New York University, New York, New York, United States of America, **3** Department of Global and Environmental Health, School of Global Public Health, New York University, New York, New York, United States of America

* tozan@nyu.edu

## Abstract

Research on health systems resilience during the Coronavirus Disease-2019 pandemic frequently used the Global Health Security Index (GHSI), a composite index scoring countries' health security and related capabilities. Conflicting results raised questions regarding the validity of the GHSI as a reliable index. This study attempted to better characterize when and to what extent countries' progress towards Global Health Security (GHS) augments health systems resilience. We used longitudinal data from 191 countries and a difference-in-difference (DiD) causal inference strategy to quantify the effect of countries' GHS capacity as measured by the GHSI on their coverage rates for essential childhood immunizations, a previously established proxy for health systems resilience. Using a sliding scale of cutoff values with step increments of one, we divided countries into treatment and control groups and determined the lowest GHSI score at which a safeguarding effect was observed. All analyses were adjusted for potential confounders. World Bank governance indicators were employed for robustness tests. While countries with overall GHSI scores of 57 and above prevented declines in childhood immunization coverage rates from 2020–2022 (coef: 0.91; 95% CI: 0.41–1.41), this safeguarding effect was strongest in 2021 (coef: 1.23; 95% CI: 0.05–2.41). Coefficient sizes for overall GHSI scores were smaller compared to several GHSI sub-components, including countries' environmental risks (coef: 4.28; 95% CI: 2.56–5.99) and emergency preparedness and response planning (coef: 1.82; 95% CI: 0.54–3.11). Our findings indicate that GHS was positively associated with health systems resilience during the pandemic (2020) and the following two years (2021–2022), that GHS may have had the most significant protective effects in 2021 as compared with 2020 and 2022, and that countries' underlying characteristics, including governance quality, bolstered health systems resilience during the pandemic.

**Data availability statement:** Our curated dataset used in this study is publicly available at https://github.com/tyh255/GHSI_2024.

**Funding:** The authors received no specific funding for this work.

**Competing interests:** The authors have declared that no competing interests exist.

## Introduction

Research on health systems resilience during the Coronavirus Disease-2019 (COVID-19) pandemic frequently focused on evaluating the role of Global Health Security (GHS), a global policy framework defined as "the existence of strong and resilient health systems that can prevent, detect, and respond to infectious disease threats" [1]. To do so, a large body of empirical research employed the Global Health Security Index (GHSI), a composite index first released in November 2019 as "the first comprehensive assessment of health security and related capabilities across the 195 countries that make up the States Parties to the International Health Regulations" [2]. The GHSI built on the Global Health Security Agenda and was developed due to concerns that the widely-used WHO Joint External Evaluation (JEE) scores were subject to self-assessment and selection biases and were unconducive to cross-country comparisons [3,4]. The GHSI comprises six categories spanning 37 indicators—which themselves are composites of 96 sub-indicators—all of which are normalized, neutrally weighted, and aggregated into an overall GHSI score [5]. Prior to the pandemic, GHSI scores were found to be associated with reduced deaths from communicable diseases [3], but some scholars questioned the GHSI's reliance on open-source information; its emphasis on indicators, such as biosecurity, that may be more relevant to high-income countries; and the overall selection process of the index's indicators [6,7].

The GHSI was nonetheless widely viewed as a transparent and veracious index and was frequently operationalized in evaluative studies of health systems performance during the COVID-19 pandemic. However, scholars alternatingly found that countries' progress towards GHS, as measured by GHSI scores, had null [8–10], mixed [11], detrimental [12], or beneficial [13,14] effects on health systems performance. These findings' inconsistencies were attributed to factors such as the GHSI's potential overestimation or underestimation of national health systems' robustness [9]; its failure to adequately account for contextual, political, public, and environmental factors [15]; or flaws in its aggregation or weighting methods [9]. These inconsistent findings may have also stemmed from methodological issues in the studies themselves and their reliance on COVID-19 outcomes. First, as the pandemic's impact extended beyond 2020, studies that only examined the relationship between GHSI scores and countries' health systems performance in 2020 might have overlooked the possible long-term protective effects of GHS capacity [16]. Second, oversimplified data analyses and the misclassification of COVID-19 deaths may have contributed to findings suggesting an inverse correlation between GHSI scores and health systems performance during the pandemic [17]. Notably, higher GHSI scores were found to be positively associated with higher completeness rates in COVID-19 infection and mortality data, which may have skewed the results [18].

In response to these contradictory results, the developers of the GHSI encouraged researchers to "examine scores at more granular levels… and consider more nuanced outcomes," including examining the relationships between health systems performance and GHSI categories rather than the overall GHSI score alone [2]. This approach opened the door for increasingly multifaceted analyses aimed at elucidating the individual drivers of health systems resilience, which we have defined for the purposes of this paper as the functioning of health systems throughout long-shocks like a pandemic [19]. However, breaking a composite index into its sub-components risks leaving out crucial systems-level dynamics and interactions [20], and may suggest that the GHSI does not necessarily measure GHS capacity as a single, unified concept.

To better understand the varying results and more accurately assess the effect of countries' progress towards GHS on health systems resilience during the pandemic, we used longitudinal data and a difference-in-difference causal inference strategy to quantify the effect of GHSI scores on coverage rates for essential childhood immunization. These rates serve as a well-established proxy for health systems resilience, as immunization is considered an essential

health service in all health care settings. Declines in coverage rates following a shock are thus considered indicative of lower levels of resilience [14,21]. Given the significance of GHS as a global policy framework, understanding its effect—if any—on health systems performance during the pandemic is critical for policymakers.

## Methods

This analysis employed longitudinal data from 191 countries and a standard quasi-experimental difference-in-difference causal inference design to quantify the effect of GHSI scores on countries' essential childhood immunization coverage rates [14,21]. There is a global consensus that the provision of key childhood vaccines constitutes an essential health service [22]. As such, this data is subject to a standardized surveillance and review process and has been consistently reported by countries to the WHO and UNICEF for decades [23]. We chose not to use mortality and morbidity rates due to COVID-19 because of concerns about their validity and bias, which compromise comparability across countries [24]. Further, these rates may be confounded by factors such as population age structures and underlying morbidities/co-morbidities [25], which limits the ability to infer causality between countries' policies and the resilience of their health systems.

### Data

Our dependent variable—national essential childhood immunization coverage estimates—was obtained from the WHO/UNICEF Joint Estimates of National Immunization Coverage [26]. This dataset includes annual vaccination coverage estimates for 14 essential childhood vaccines across 195 countries from 1997 to 2022. For our analysis, we used data from 2015 to 2022 that met the required parallel pre-trend assumption (S2 Fig) [14]. In accordance with established methodological guidelines on data imbalance and availability [14,21], we excluded the yellow fever vaccine (YFV) from our analysis, as it is not widely administered across all countries, leading to data imbalance [21]. We also excluded the first dose of the inactivated polio vaccine (IPV-1) due to the unavailability of IPV-1 data across all years under observation [21]. Following these exclusions, our analysis included 12 different childhood vaccines: Bacille Calmette-Guérin (BCG); the first and third dose of diphtheria, tetanus toxoid, and pertussis containing vaccine (DTP1, DTP3); the birth dose of hepatitis B vaccine (HEPB-3); the third dose of hepatitis B containing vaccine (HEPBB); the third dose of *Haemophilus influenzae* type B containing vaccine (HIB3); the first and second doses of measles containing vaccine (MCV1, MCV2); the third dose of pneumococcal conjugate vaccine (PCV3); the third dose of polio containing vaccine (POL3); the second or third dose of rotavirus vaccine (ROTAC); and the first dose of rubella containing vaccine (RCV1). Most of these vaccines were for infants for whom coverage was estimated during the first year of life, but some vaccines, such as MCV2, were for older children with coverage rates estimated during the second year of life. A list of recommended ages for estimating vaccination coverage is available in the WHO/UNICEF Joint Estimates of National Immunization Coverage methodology [26]. A number of countries—Austria, Czechia, Denmark, Finland, France, Greece, Ireland, Israel, Italy, Malta, Portugal, and Slovenia—had missing data on BCG vaccinations for 2022 or did not report data due to variations in national vaccination requirements. These missing observations accounted for just 0.07% of our total sample.

Our primary exposure variable was the 2019 GHSI scores for each country, released in 2021 following a methodological update for the second iteration of the index [5]. This dataset consists of six GHSI categories—prevention, detection and reporting, rapid response, health system, compliance with international norms, and risk environment—measured by a combination of 37 indicators, which are themselves composites of 96 sub-indicators (S1 Table). These sub-indicators, indicators, and categories are subsequently normalized, equally

weighted, and then aggregated into an overall GHSI score [5]. The GHSI and its six categories range from 0 to 100, with 100 representing the highest level of preparedness capacity, both overall and in each individual category. We re-computed the overall GHSI score and its first category (countries' ability to prevent the emergence or release of pathogens) after removing the vaccination coverage indicator and re-averaging the remaining indicators with equal weights. This adjustment ensured that our dependent and independent variables were not drawn from the same data.

Our dataset also included the World Bank's income level data (2019), which we used to categorize countries into high, upper-middle, lower-middle, or low income groups, as well as the WHO's regional classifications [27,28]. For robustness checks, we repeated our analyses using the World Bank's Governance Indicators (2019), which include indices for governance effectiveness, rule of law, and control of corruption [29]. These indicators ranged from −2.5 to 2.5, with 2.5 representing the greatest progress towards the theme of each index (e.g., effective governance). Countries with significant missing data, specifically the Cook Islands, Niue, the State of Palestine, and the Democratic People's Republic of Korea, were excluded from the analysis. As a result, our sample included data from 191 countries.

## Statistical analysis

We employed the doubly-robust DiD estimation methods proposed by Sant'Anna and Zhao [30] to obtain the average treatment effect on the treated (ATT) of higher scores of 36 GHSI indicators, six GHSI categories, and the overall GHSI in safeguarding essential childhood immunization coverage during the COVID-19 pandemic (2020) and the following two years (2021–2022). This DiD method offered two primary advantages: first, it allowed us to systematically test our parallel trend assumptions; and second, in contrast to other DiD estimators, Sant'Anna and Zhao's estimators are less sensitive to model misspecifications. Our outcome model is enumerated in Equation 1, wherein the binary *Treated* variable represents countries' assigned group based on their GHSI scores (for instance, based on the below criteria, we assigned countries in our main model to high and low GHSI groups based on a cutoff score of 57). We included time (year) and country fixed effects (as represented respectively by $\gamma_t$ and $\delta_i$), $\theta$ is the coefficient for the interaction term $\text{Treated}_i \times \text{Post}_t$, and $X_{it}^{\top}$ is a vector of covariates including region and income group. The doubly-robust DiD estimator works by combining the outcome model with a treatment model (handled by the interaction term), adjusting for covariates and time-varying factors to provide a consistent estimate of the treatment effect $\theta$. The doubly-robust DiD estimator combines two components: inverse probability of treatment weighting (IPTW), which is estimated from a propensity score model, and outcome model adjustment. This approach allows for appropriate adjustment of covariates and time-varying factors, yielding a consistent estimate of the treatment effect. By including the same covariates in both the propensity score model and the outcome model, this method protects against model misspecification and enhances the robustness of the treatment effect estimate.

$$Coverage_{it} = \alpha + \gamma_t + \delta_i + \boldsymbol{\theta}\big(\text{Treated}_i \times \text{Post}_t\big) + X_{it}^{\top}\beta + \epsilon_{it} \qquad \text{(Equation 1)}$$

The onset of the COVID-19 pandemic was used to define the pre-post period, wherein the years prior to 2020 were defined as pre-pandemic and 2020–2022 were defined as post-pandemic. For the overall GHSI score, its six categories, and its 36 indices, we divided countries into treatment and control groups by testing cutoff values on a sliding scale from 0 to 100 with step increments of one. This process allowed us to determine the minimum GHSI score at which the parallel pre-trend assumption was met and a positive protective effect was observed, indicating countries' progress towards that policy goal was sufficient to yield

a significant safeguarding effect [14,21]. For robustness, we tested the effect of countries' progress along the World Bank's Governance Indicators (2019) on countries' essential childhood immunization coverage rates to help validate our governance-related findings. While the World Bank's Government Effectiveness indicator (one of the three World Bank indicators we tested) generally measures similar thematic areas as the GHSI's government effectiveness (6.1.1) and political and security risk (6.1) categories, we assessed that the differences were large enough to be meaningful, with correlations of 0.91 and 0.74, respectively. We included countries' income group as per the World Bank classification and countries' geographical region as per the WHO regional classification as covariates in our analyses. All analyses were conducted using R software (Version 4.0.3). The reporting of this study conformed with the STROBE guidelines (S1 Checklist). Our curated dataset used in this study is publicly available at https://github.com/tyh255/GHSI_2024.

### Patient and public involvement

Patients and/or the public were not involved in the design, or conduct, or reporting, or dissemination plans of this research.

## Results

### Descriptive analysis

Fig 1 and Table 1 show countries included in the analysis and their summary statistics stratified by GHSI scores, respectively. Additional details about the dataset can be found in S1 Text. The

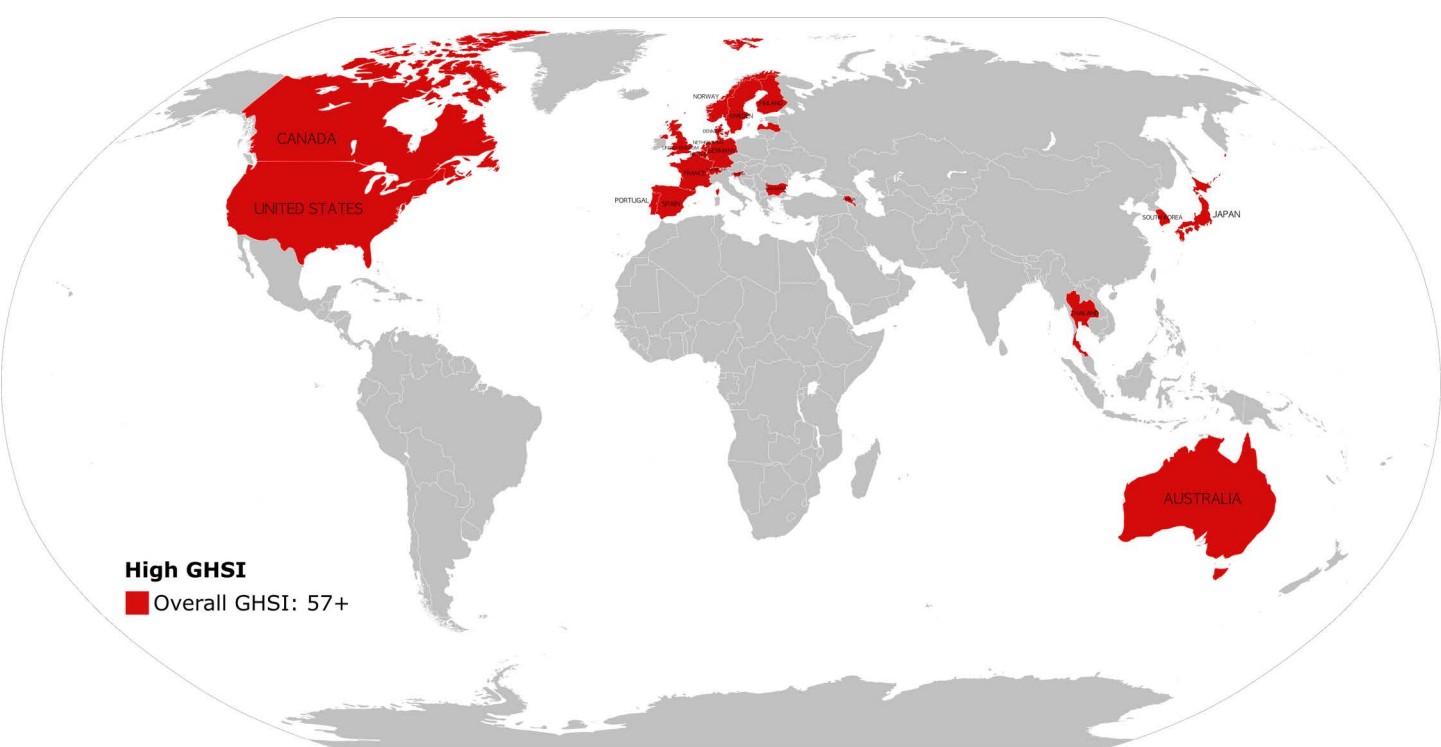

**Fig 1. Map of countries with high Global Health Security capacity (GHSI ≥ 57).** *Countries which had an overall GHSI score of or above 57 included Armenia, Australia, Belgium, Bulgaria, Canada, Denmark, Finland, France, Germany, Japan, Latvia, Netherlands, Norway, Portugal, Republic of Korea, Slovenia, Spain, Sweden, Switzerland, Thailand, United Kingdom, and the United States. A full list of all countries is included in S2 Table. The public domain base map is available at https://commons.wikimedia.org/wiki/File:CIA_WorldFactBook-Political_world.svg and is derived from the CIA World Factbook. This map is for illustrative purposes only.*

**Table 1. Childhood immunization and summary statistics by GHSI scores (2019).**

| | Countries with GHSI ≥ 57 (N = 22) | Countries with GHSI < 57 (N = 169) | Total (N = 191) |
|---|---|---|---|
| GHSI Mean (SD) | 64.6 (5.0) | 34.4 (10.2) | 37.8 (13.6) |
| World Bank Income Group (N, %) | | | |
| High | 19 (31.7) | 41 (68.3) | 60 |
| Upper-Middle | 3 (5.6) | 51 (94.4) | 54 |
| Lower-Middle | 0 (0.0) | 48 (100.0) | 48 |
| Low | 0 (0.0) | 29 (100.0) | 29 |
| WHO Region (N, %) | | | |
| Americas | 2 (5.7) | 33 (94.3) | 35 |
| Europe | 16 (30.2) | 37 (69.8) | 53 |
| Western Pacific | 3 (12.0) | 22 (88.0) | 25 |
| Eastern Mediterranean | 0 (0.0) | 21 (100.0) | 21 |
| Southeast Asia | 1 (10.0) | 9 (90.0) | 10 |
| Africa | 0 (0.0) | 47 (100.0) | 47 |
| Immunization Coverage Rates (Mean, SD) | | | |
| 2015–2019 | 92.9 (8.7) | 86.3 (15.8) | 87.0 (15.3) |
| 2020 | 92.8 (8.4) | 83.7 (15.4) | 84.7 (15.0) |
| 2021 | 92.2 (8.5) | 82.2 (17.1) | 83.3 (16.7) |
| 2022 | 92.4 (8.7) | 83.1 (16.7) | 84.1 (16.3) |
| 2015–2022 | 92.8 (8.6) | 85.0 (16.1) | 85.9 (15.7) |

minimum treatment cutoff value at which a significant protective effect was first observed was 57 (89th percentile) for the overall GHSI score; 67 (97th percentile) for the prevention of the emergence or release of pathogens; 55 (96th percentile) for the early detection and reporting of epidemics of potential international control; 65 (94th percentile) for the rapid response and mitigation of the spread of an epidemic; 55 (89th percentile) for sufficient and robust health systems to treat the sick and protect health workers; 69 (95th percentile) for commitments to improving national capacity, financing plans to address gaps, and adherence to global norms; and 70 (82nd percentile) for countries' overall risk environment and vulnerabilities to biological threats (Fig 3). Vaccination rates were generally lower from 2020–2022 compared to 2015–2019. As shown in Fig 2A , this decline was particularly pronounced for countries with GHSI scores below 57, where the average coverage dropped from 86.3% prior to the pandemic (2015–2019) to 83.7% in 2020, 82.2% in 2021, and 83.1% in 2022. In contrast, countries with GHSI scores of 57 and above experienced a smaller decline, with average coverage dropping from 92.9% prior to the pandemic (2015–2019) to 92.8% in 2020, 92.2% in 2021, and 92.4% in 2022.

## Overall GHSI and sub-component analyses

Table 2 presents the results of the DiD models analyzing the relationship between countries' overall GHSI and individual GHSI category scores and childhood immunization coverage rates. From 2020 to 2022, the relationship between overall GHSI scores and coverage rates was positive and statistically significant (average coefficient: 0.91; 95% CI: 0.41–1.41). In other words, we found that countries with an overall GHSI score of 57 and above prevented an average 0.91 percentage point decline in vaccination coverage each year from 2020 to 2022. As shown in Fig 2B, this positive coefficient size was greater in 2021 (average coefficient: 1.23; 95% CI: 0.05–2.41) as compared to 2020 (average coefficient: 0.74; 95% CI: 0.28–1.20) and to 2022 (average coefficient: 0.77; 95% CI: 0.06–1.46). As presented in Table 2, the effect sizes

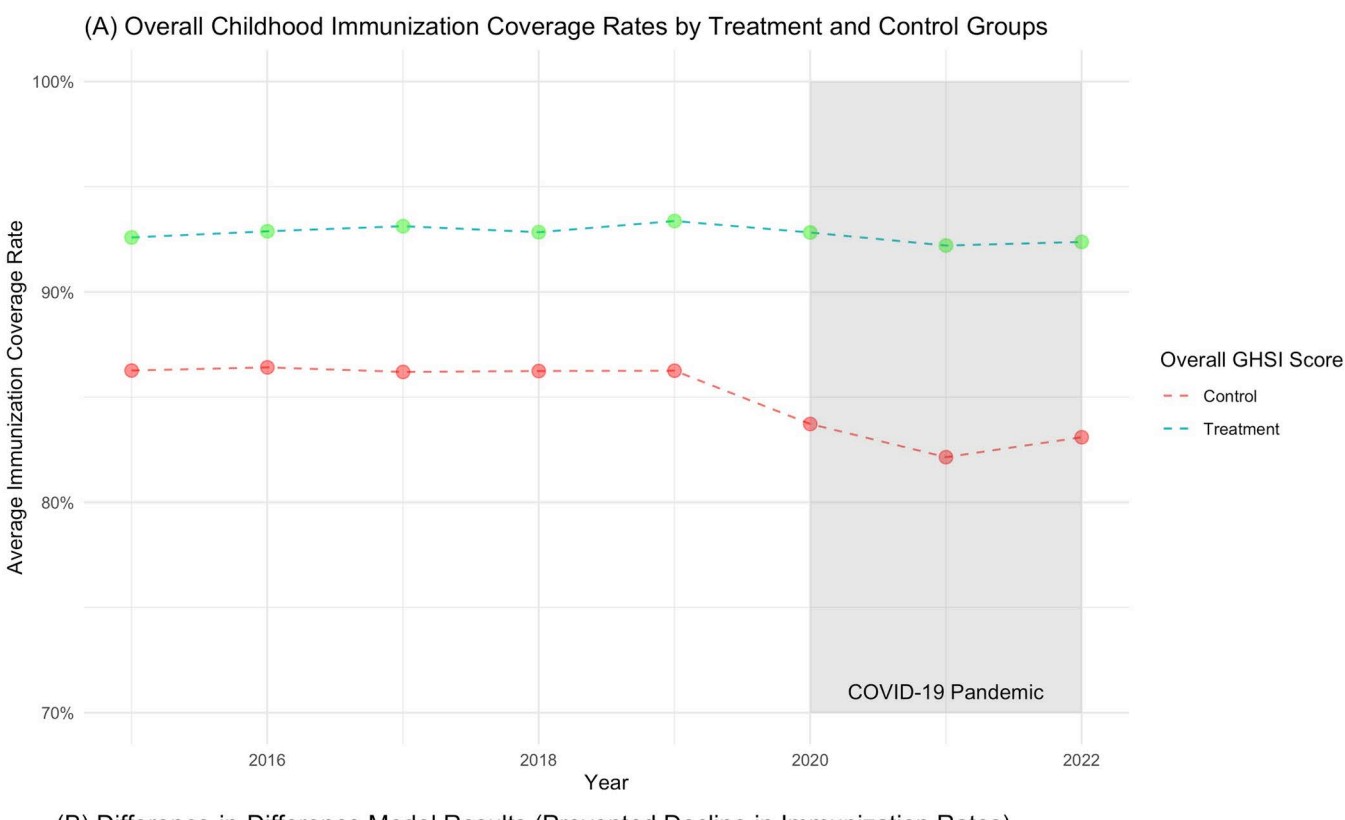

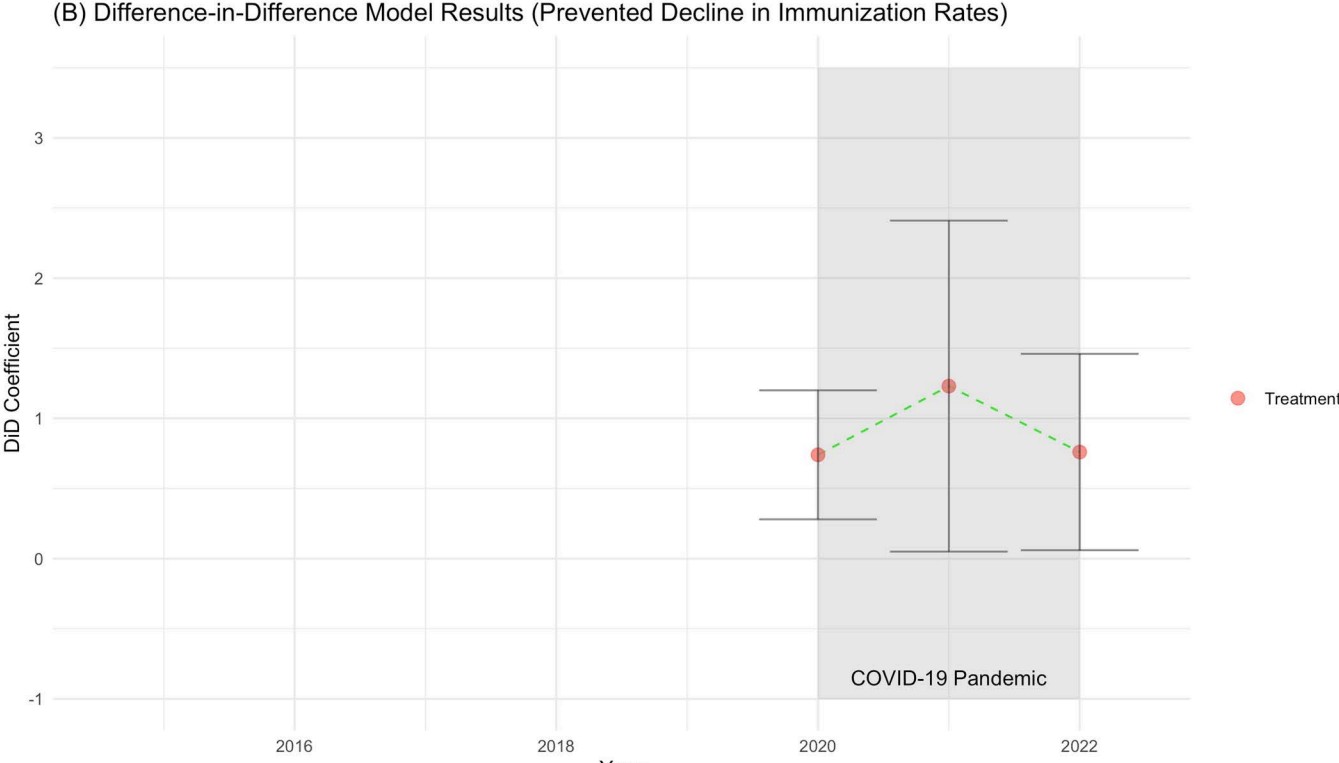

**Fig 2. Average childhood immunization coverage rates by treatment (GHSI $\geq 57$) and control (GHSI < 57) groups and overall effect of treatment (GHSI $\geq 57$) on preventing declines in childhood immunization coverage rates with 95% confidence intervals.**

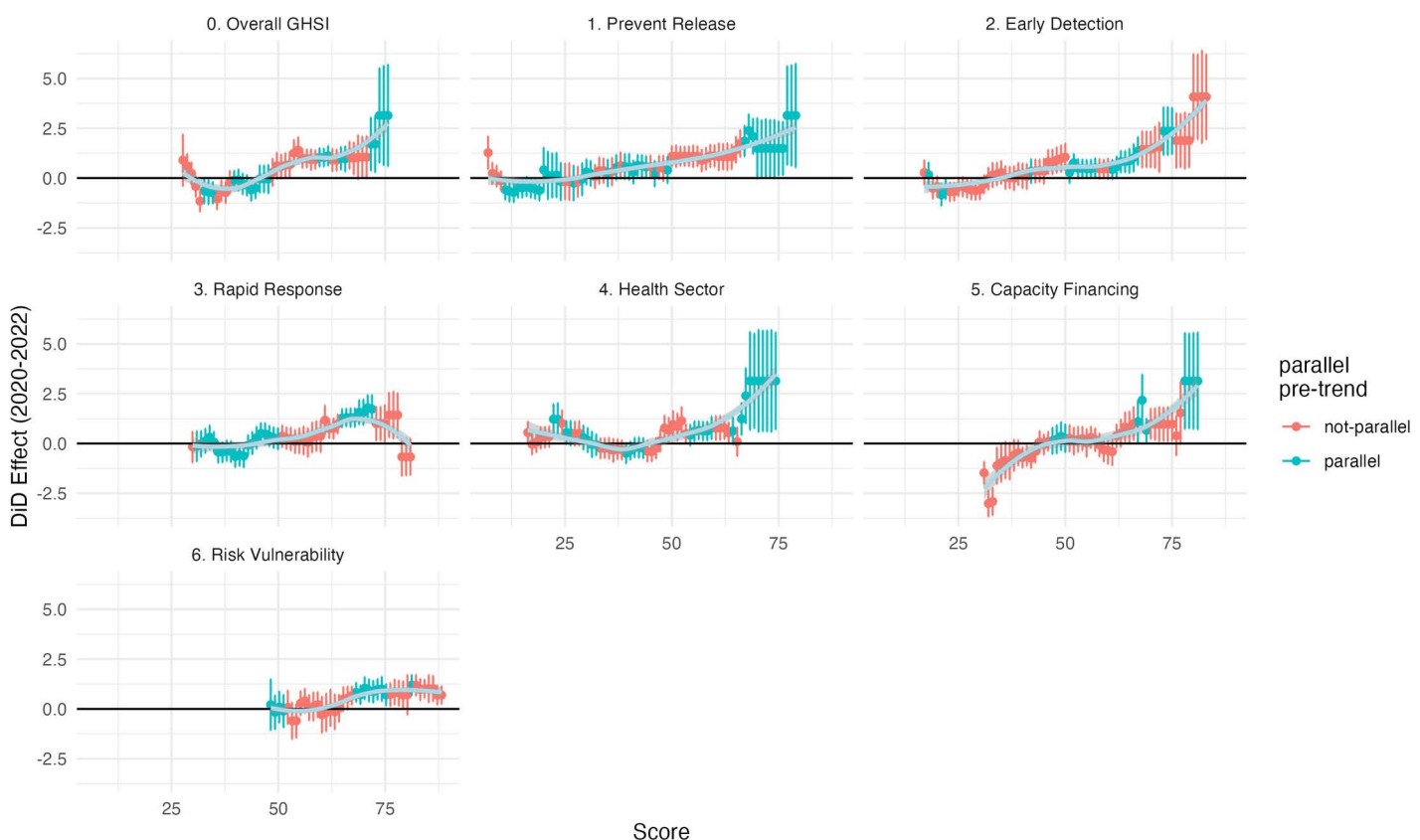

**Fig 3. Difference-in-difference model results for overall GHSI and GHSI categories (2020–2022).** *DiD models were only run when there were pre-treatment periods to test. Trendlines and their standard errors (in light blue) were calculated based on local polynomial regression fitting.*

**Table 2. Difference-in-difference model results for overall GHSI and GHSI categories (2020–2022).**

| | Average DiD effect size (2020–2022) | DiD effect size for 2020 | DiD effect size for 2021 | DiD effect size for 2022 | *p-value* for parallel trend | Cutoff value (percentile) |
|---|---|---|---|---|---|---|
| Overall GHSI Scores (95% CI) | 0.91* (0.41, 1.41) | 0.74* (0.28, 1.20) | 1.23* (0.05, 2.41) | 0.76* (0.06, 1.46) | 0.11 | 57 (0.89) |
| 1: Prevention of the Emergence or Release of Pathogens | 1.86* (1.05, 2.66) | 1.45* (0.61, 2.30) | 2.74* (0.77, 3.72) | 1.37* (0.21, 2.52) | 0.33 | 67 (0.97) |
| 2: Early Detection and Reporting for Epidemics of Potential International Concern | 0.82* (0.24, 1.40) | 1.02* (0.46, 1.57) | 1.17 (−0.03, 2.37) | 0.28 (−0.53, 1.09) | 0.53 | 55 (0.96) |
| 3: Rapid Response to and Mitigation of the Spread of an Epidemic | 1.30* (0.83, 1.78) | 0.84* (0.34, 1.34) | 2.04* (1.06, 3.03) | 1.02* (0.34, 1.69) | 0.28 | 65 (0.94) |
| 4: Sufficient and Robust Health System to Treat the Sick and Protect Health Workers | 0.42 (−0.12, 0.96) | −0.09 (−0.74, 0.56) | 0.98* (0.03, 1.99) | 0.37 (−0.44, 1.17) | 0.11 | 55 (0.89) |
| 5: Commitments to Improving National Capacity, Financing Plans to Address Gaps, and Adhering to Global Norms | 0.66* (0.02, 1.30) | 0.63 (−0.03, 1.29) | 0.93 (−0.12, 1.97) | 0.43 (−0.41, 1.26) | 0.05 | 69 (0.95) |
| 6: Overall Risk Environment and Country Vulnerability to Biological Threats | 1.04* (0.44, 1.64) | 1.07* (0.41, 1.72) | 0.96 (−0.08, 1.99) | 1.10* (0.28, 1.93) | 0.05 | 70 (0.82) |

* indicates statistical significance (95% CI).

varied across the six GHSI categories, with the strongest results associated with the "prevention of the emergence or release of pathogens" category (average coefficient: 1.86; 95% CI: 1.11–2.60) and the "rapid response to and mitigation of the spread of an epidemic" category (average coefficient: 1.30; 95% CI: 0.83–1.78).

We found that across most GHSI categories, the results in 2021 were stronger than in 2020 or 2022. The exception was the sixth GHSI category—countries' overall risk environment and vulnerability to biological threats—where the coefficient effect size in 2021 (average coefficient: 0.96; 95% CI: −0.08–1.99) was smaller compared to 2020 (average coefficient: 1.07; 95% CI: 0.41–1.72) and to 2022 (average coefficient: 1.10; 95% CI: 0.28–1.93). This category was also notable for having the lowest treatment cutoff percentile (82nd percentile) at which a significant protective effect was observed.

We also observed a general trend, as shown in Fig 3, where the DiD effect sizes became more significant as the treatment cutoff increased. For instance, at the cutoff value of 65 (95th percentile), the relationship between overall GHSI scores and childhood immunization rates was positive and statistically significant (average coefficient: 0.97; 95% CI: 0.37–1.57) from 2020–2022. However, at the extreme end of the scale (cutoff: 74; 99th percentile), the effect size was significantly higher (average coefficient: 3.14; 95% CI: 0.62–5.66). This trend was most pronounced for the fourth category, which assesses the sufficiency and robustness of health systems to treat the sick and protect health workers. The average coefficient from 2020–2022 was 0.42 (95% CI: −0.12–0.96) at the 89th percentile, but at the 99th percentile (cutoff: 68), the coefficient increased to 2.39 (95% CI: 1.01–3.78). However, this trend of increasing DiD effect sizes with higher overall GHSI and GHSI category scores was not observed for the "rapid response to and mitigation of epidemics" category where higher values failed to meet the parallel pre-trend assumption and even showed negative effects. Similarly, the "risk environment and vulnerability" category exhibited a modest and yet significant positive effect at lower cutoffs, but this effect quickly plateaued at higher cutoffs.

## GHSI indicator analyses

We next analyzed the potential protective effects of the 36 GHSI indicators on essential childhood immunization coverage during the pandemic. The most significant results are reported in Table 3, with full results available in S3–S8 Figs and S3–S16 Tables. From 2020 to 2022, we found that the most significant indicators were "environmental risks" (average coefficient: 4.28; 95% CI: 2.56–5.99), "biosecurity" (average coefficient: 1.87; 95% CI: 0.83–2.91), and "emergency preparedness and response" (average coefficient: 1.82; 95% CI: 0.54–3.11). Consistent with previous findings, the DiD effect sizes in 2021 were generally larger than in 2020 or 2022. However, there were to two notable exceptions. First, the effects of countries' supply chains for health systems and healthcare workers showed a stronger DiD effect in 2022 (average coefficient: 1.74; 95% CI: 0.84–2.64) compared to 2021 (average coefficient: 1.29; 95% CI: 0.08–2.49) and to 2020 (average coefficient: −0.45; 95% CI: −1.87–0.98). Second, the DiD effect for countries' public health vulnerabilities was greater in 2022 (average coefficient: 2.44; 95% CI: 0.88–3.99) than in 2021 (average coefficient: 1.58; 95% CI: 0.06–3.09) and 2020 (average coefficient: 0.51; 95% CI: −0.39–1.42).

## Robustness analysis

Given that our GHSI indicator findings suggested that countries' underlying characteristics, such as infrastructure adequacy and governance quality, were significant determinants of health systems resilience, we tested alternate indicators from the World Bank to confirm our findings (Table 3). From 2020–2022, both countries' governance effectiveness (average coefficient: 1.04; 95% CI: 0.52–1.55) and rule of law (average coefficient: 1.08; 95% CI: 0.49–1.66)

**Table 3. Difference-in-difference model results for significant GHSI Indicators and World Bank Indices (2020–2022).**

| Category-Subcategory | Average DiD effect size (2020–2022) | DiD effect size for 2020 | DiD effect size for 2021 | DiD effect size for 2022 | p-value for parallel trend | Cutoff value (percentile) |
|---|---|---|---|---|---|---|
| GHSI 1.3: Biosecurity | 1.87* (0.83, 2.91) | 1.25* (0.18, 2.31) | 3.25* (1.87, 4.62) | 1.12 (−0.51, 2.74) | 0.44 | 77 (0.98) |
| GHSI 3.1: Emergency Preparedness and Response Planning | 1.82* (0.54, 3.11) | 1.45* (0.06, 2.84) | 2.94* (0.11, 5.77) | 1.08* (0.12, 2.05) | 0.06 | 84 (0.98) |
| GHSI 4.1 Health capacity in clinics, hospitals, and community care centers | 1.14* (0.68, 1.59) | 0.79* (0.25, 1.35) | 1.59* (0.79, 2.39) | 1.02* (0.24, 1.81) | 0.31 | 54 (0.96) |
| GHSI 4.2: Supply chain for health system and healthcare workers | 0.86* (0.09, 1.63) | −0.45 (−1.87, 0.98) | 1.29* (0.08, 2.49) | 1.74* (0.84, 2.64) | 0.33 | 73 (0.98) |
| GHSI 5.3: International Commitments | 1.27* (0.58, 1.97) | 0.42 (−0.47, 1.31) | 1.79* (0.48, 3.09) | 1.61* (0.66, 2.56) | 0.32 | 97 (0.88) |
| GHSI 6.2: Socioeconomic Resilience | 0.61* (0.05, 1.16) | 0.48* (0.09, 0.86) | 1.48* (0.31, 2.66) | −0.13 (−1.01, 0.74) | 0.21 | 94 (0.94) |
| GHSI 6.3: Infrastructure Adequacy | 0.79* (0.23, 1.36) | 0.64* (0.05, 1.22) | 0.89 (−0.34, 2.13) | 0.86* (0.15, 1.57) | 0.12 | 76 (0.87) |
| GHSI 6.4: Environmental Risks | 4.28* (2.56, 5.99) | 3.04* (1.53, 4.54) | 6.84* (4.65, 9.03) | 2.96 (−1.41, 7.33) | 0.19 | 72 (0.96) |
| GHSI 6.5: Public Health Vulnerabilities | 1.51* (0.61, 2.41) | 0.51 (−0.39, 1.42) | 1.58* (0.06, 3.09) | 2.44* (0.88, 3.99) | 0.43 | 51 (0.38) |
| World Bank: Governance Effectiveness | 1.04* (0.52, 1.55) | 0.54* (0.10, 0.97) | 1.62* (0.65, 2.59) | 0.95* (0.19, 1.70) | 0.09 | 1.7 (0.94) |
| World Bank: Rule of Law | 1.08* (0.49, 1.66) | 0.80* (0.36, 1.24) | 1.17 (−0.46, 2.79) | 1.26* (0.49, 2.02) | 0.07 | 1.3 (0.88) |

* indicates statistical significance (95% CI). World Bank Index Range: (−2.5–2.5). The Control of Corruption variable did not satisfy the parallel pre-trend assumption (i.e., the treatment and control groups did not have parallel vaccination rates prior to 2020) necessary for a difference-in-difference test, and therefore was not included in the results.

were both statistically significant and carried similar substantive significance. In contrast, countries' control of corruption did not satisfy the parallel pre-trend assumption. The DiD effect size for governance effectiveness was greatest in 2021 (average coefficient: 1.62; 95% CI: 0.65–2.59) whereas the DiD effect size for rule of law was largest in 2022 (average coefficient: 1.26; 95% CI: 0.49–2.02).

## Discussion

Our analyses showed that GHS played a significant role in safeguarding health systems resilience during the COVID-19 pandemic. We found evidence suggesting that countries with high GHS capacity (GHSI ≥ 57) safeguarded against an average annual 0.91% decline in essential childhood immunization coverage rates from 2020 to 2022. Specifically, countries with less progress towards GHS (GHSI < 57) saw larger declines in immunization rates during the pandemic, averaging a 4.1 percentage point decline in coverage rates in 2021 compared to 2015–2019. In contrast, countries with high GHS capacity experienced a smaller decline, with an average drop of 0.7 percentage points from their pre-pandemic immunization rates. We also found that the GHSI's categories and sub-categories often had more significant safe-guarding effects than the overall GHSI score. However, the relatively high cutoff values for all categories and sub-categories, as shown in Tables 2 and 3, suggest that countries must make substantial progress in these areas to reap the protective benefits during a health shock. Some of the most significant sub-components included "environmental risks," "biosecurity," and "emergency preparedness and response planning." The large effect size of environmental risks

is particularly noteworthy. Environmental risk is a composite measure, incorporating factors such as urbanization (greater urbanization leads to lower scores), land use (more deforestation leads to lower scores), and natural disaster risk (higher disaster risks leads to lower scores due to potential negative risk to economic circumstances). The large effect size may suggest a possible mechanism: evidence indicates that more urbanized countries experienced higher COVID-19 incidence rates [31]. This higher burden of disease may have, in turn, negatively affected health systems resilience. The mechanism behind the relationship between emergency preparedness and health systems resilience appears more straightforward. Countries that had prepared for national emergencies were comparatively better equipped to respond to the COVID-19 pandemic. As a result, they could allocate fewer resources to emergency responses and more resources to other essential health systems functions, such as vaccination programs, than countries that were less prepared.

This study is among the first to examine the relationship between GHSI and health systems resilience from 2020 to 2022. We identified a general increase in coefficient sizes from 2020 to 2021, followed by a decline from 2021 to 2022. This temporal variation supports the hypothesis that countries' progress towards GHS increases resilience to health shocks such as the COVID-19 pandemic. As the burden of disease peaked in 2021, GHS' protective effects on childhood immunization coverage, a proxy for overall health systems resilience, was also at its highest. One possible mechanism is that progress towards GHS strengthens health system infrastructure and capacity, enabling critical health system functions to be maintained during a health shock [32]. Another possibility is that countries with more progress towards GHS may have had more capacity, especially circa 2021, to adopt to the COVID-19 emergency measures and then reallocate resources to sustain routine health services later in the year. In contrast, countries with less progress towards GHS struggled to ramp up and then scale down their COVID-19 programs [32]. Despite a 0.8 percentage point increase in the average immunization coverage rate from 2021 to 2022, the average coverage rate of 84.1% in 2022 remains significantly lower than the global average of 87.0% from 2015–2019 (t = −7.34; *p-value* < 0.001). Further research and data on subsequent years is needed to more assess whether this general decline in effect sizes persists and, if so, to understand the underlying causes.

When examining the relationship between GHSI sub-components and essential childhood immunization coverage, we generally found that the most substantively and statistically significant indicators were often proxies for countries' underlying characteristics (like the share of population in urban settings) and governance quality. The causal mechanism for the positive effect of governance quality on health systems resilience is likely straightforward: countries with effective governance are better able to appropriately allocate resources, manage logistical challenges, and adapt to changing circumstances. These characteristics enable them to efficiently deploy resources to maintain both COVID-19 emergency measures and non-COVID health services [32]. To confirm this effect, we tested the World Bank's governance indicators, finding that both governance effectiveness and the rule of law were positively associated with safeguarding against declines in essential childhood immunization coverage. One unexpected finding from this robustness analysis was that while the effect of governance effectiveness on health systems resilience was more significant in 2021 than 2022, similar to the general trend with the overall GHSI scores, the inverse was true for the rule of law, where the effect size was both larger and more significant in 2022 than in 2021 or 2020. This may suggest that the rule of law is positively related to continued adherence to essential service delivery standards by government and frontline workers [33], which could have contributed to larger effect sizes after 2020 as countries stabilized after the initial COVID-19 shock. However, further research is needed to fully understand the effects of governance effectiveness and the rule of law on health systems resilience, both during and after the pandemic.

The primary strength of this study lies in its application of best practices for evaluating the relationship between GHS and health systems resilience, building on previous methodological considerations in the literature. Specifically, we used longitudinal data, employed a robust causal inference methodology, and disaggregated the GHSI into its sub-components. Our findings align with more recent research suggesting that progress towards GHS positively impacts health systems resilience during the pandemic [14,16,21]. While our overall findings were similar to those of these studies, slight differences in effect sizes were likely due to the use of different statistical methods and cutoffs. For instance, Kim (2023) defined high GHSI scores as those in the 80th percentile (a score of 51) and found that countries with high scores prevented a 2.02% reduction in immunization coverage in 2021, with no significant effect in 2020 [14]. In contrast, this study used a slightly higher cutoff value of 57, determined by a more robust methodology, and found that countries with high GHSI scores prevented a 0.74% decline in 2020, a 1.23% decline in 2021, and a 0.76% decline in 2022. Our strong findings at the sub-category level further corroborate previous suggestions that disaggregating the GHSI enhances analyses of health systems resilience during the pandemic [2] and that the relationship between GHSI and health systems resilience is most pronounced over a multi-year time horizon [16]. As noted in the literature, measuring a multi-dimensional concept like GHS is challenging due to methodological issues related to weighting and aggregation processes and the inherently subjective procedure of creating composite indices [20]. While disaggregating GHSI into its sub-components helps address concerns related to index weighting and aggregation by clarifying the impact of individual factors, this approach may limit the ability to evaluate GHS as a unified policy framework.

This study is not without limitations. First, our results suggest that there are intricate feedback effects at play among components of this complex system, as evidenced by the significant and time-dependent variations in GHSI indicator coefficients. Further research using advanced analytical methodologies is needed to unravel these dynamics, which are important to understand the long-term impacts of decreased childhood immunization coverage rates and identify the key drivers of health systems resilience during public health crises. Second, this study used a single indicator—childhood immunization coverage rates—as a proxy for health systems resilience during the pandemic. While this choice offers advantages over using direct measures like COVID-19 mortality rates (as was noted earlier), future research should incorporate additional proxy indicators to enhance robustness. Third, this study focused on the years from 2020–2022 as its observation period. Future studies should include longitudinal data beyond 2022 to test whether the general decline in effect sizes from 2021 to 2022 persists into 2023 and beyond to improve our understanding of the trends and their implications.

## Supporting information

**S1 Checklist. STROBE statement—checklist of items that should be included in reports of cross-sectional studies.**
(DOCX)

**S1 Text. Description of the dataset.**
(DOCX)

**S1 Table. GHSI categories and indicators.**
(DOCX)

**S2 Table. GHSI countries grouped by recomputed GHSI scores.**
(DOCX)

**S3 Table. Difference-in-difference model results for overall GHSI and GHSI categories by cutoff values (2020–2022).**
(DOCX)

**S4 Table. Difference-in-difference model results by year for overall GHSI and GHSI categories which fulfilled the parallel pre-trend assumption at varying cutoff intervals of five (2020–2022).**
(DOCX)

**S5 Table. Difference-in-difference model results for GHSI category 1 (Prevention) by cutoff values (2020–2022).**
(DOCX)

**S6 Table. Difference-in-difference model results by year for GHSI category 1 (Prevention) scores which fulfilled the parallel pre-trend assumption at cutoff intervals varying by five (2020–2022).**
(DOCX)

**S7 Table. Difference-in-difference model results for GHSI category 2 (Early Detection) by cutoff values (2020–2022).**
(DOCX)

**S8 Table. Difference-in-difference model results by year for GHSI category 2 (Early Detection) scores which fulfilled the parallel pre-trend assumption at cutoff intervals varying by five (2020–2022).**
(DOCX)

**S9 Table. Difference-in-difference model results for GHSI category 3 (Rapid Response) by cutoff values (2020–2022).**
(DOCX)

**S10 Table. Difference-in-difference model results by year for GHSI category 3 (Rapid Response) scores which fulfilled the parallel pre-trend assumption at cutoff intervals varying by five (2020–2022).**
(DOCX)

**S11 Table. Difference-in-difference model results for GHSI category 4 (Health System) by cutoff values (2020–2022).**
(DOCX)

**S12 Table. Difference-in-difference model results by year for GHSI category 4 (Health System) scores which fulfilled the parallel pre-trend assumption at cutoff intervals varying by five (2020–2022).**
(DOCX)

**S13 Table. Difference-in-difference model results for GHSI category 5 (Compliance with International Norms) by cutoff values (2020–2022).**
(DOCX)

**S14 Table. Difference-in-difference model results by year for GHSI category 5 (Compliance with International Norms) scores which fulfilled the parallel pre-trend assumption at cutoff intervals varying by five (2020–2022).**
(DOCX)

**S15 Table. Difference-in-difference model results for GHSI category 6 (Risk Environment) by cutoff values (2020–2022).**
(DOCX)

**S16 Table.** Difference-in-difference model results by year for GHSI category 6 (Risk Environment) scores which fulfilled the parallel pre-trend assumption at cutoff intervals varying by five (2020–2022).
(DOCX)

**S17 Table.** Difference-in-difference model results for World Bank Governance Indexes (Effective Governance, Rule of Law, Control of Corruption) by cutoff values (2020–2022).
(DOCX)

**S18 Table.** Difference-in-difference model results by year for World Bank Governance Index (Effective Governance, Rule of Law, Control of Corruption) scores which fulfilled the parallel pre-trend assumption at cutoff intervals varying by five (2020–2022).
(DOCX)

**S1 Fig.** Distribution of original and recomputed overall Global Health Security Index (2019) scores.
(DOCX)

**S2 Fig.** Parallel pre-trends between treatment and control groups for overall GHSI scores and GHSI categories (2015–2019).
(DOCX)

**S3 Fig.** Distribution of original and recomputed category 1 (Prevention) scores, 2019.
(DOCX)

**S4 Fig.** Difference-in-difference model results for GHSI category 1 (Prevention) (2020–2022).
(DOCX)

**S5 Fig.** Difference-in-difference model results for GHSI category 2 (Detection and Reporting) (2020–2022).
(DOCX)

**S6 Fig.** Difference-in-difference model results for GHSI category 3 (Rapid Response) (2020–2022).
(DOCX)

**S7 Fig.** Difference-in-difference model results for GHSI category 4 (Health System) (2020–2022).
(DOCX)

**S8 Fig.** Difference-in-difference model results for GHSI category 5 (Compliance with International Norms) (2020–2022).
(DOCX)

**S9 Fig.** Difference-in-difference model results for GHSI category 6 (Risk Environment) (2020–2022).
(DOCX)

**S10 Fig.** Difference-in-difference model results for World Bank Governance Indexes (Effective Governance, Rule of Law, Control of Corruption) (2020–2022).
(DOCX)

## Author contributions

**Conceptualization:** Tyler Y. Headley, Sooyoung Kim, Yesim Tozan.

**Data curation:** Tyler Y. Headley, Sooyoung Kim, Yesim Tozan.

**Formal analysis:** Tyler Y. Headley, Sooyoung Kim, Yesim Tozan.

**Investigation:** Tyler Y. Headley, Sooyoung Kim, Yesim Tozan.

**Methodology:** Tyler Y. Headley, Sooyoung Kim, Yesim Tozan.

**Supervision:** Yesim Tozan.

**Validation:** Tyler Y. Headley, Sooyoung Kim, Yesim Tozan.

**Visualization:** Tyler Y. Headley, Sooyoung Kim, Yesim Tozan.

**Writing – original draft:** Tyler Y. Headley, Yesim Tozan.

**Writing – review & editing:** Tyler Y. Headley, Sooyoung Kim, Yesim Tozan.

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
