## [Decision Letter · Decision Letter 0]

20 Sep 2024

PGPH-D-24-01417

Countries’ Progress Towards Global Health Security (GHS) Increased Health Systems Resilience During the Coronavirus Disease-19 (COVID-19) Pandemic: A Difference-in-Difference Study of 191 Countries

Dear Dr. Tozan,

Thank you for submitting your manuscript to PLOS Global Public Health. After careful consideration, we feel that it has merit but does not fully meet PLOS Global Public Health’s publication criteria as it currently stands. Therefore, we invite you to submit a revised version of the manuscript that addresses the points raised during the review process.

We look forward to receiving your revised manuscript.

Kind regards,

Sanjana Mukherjee, Ph.D., M.Sc.

Guest Editor

Journal Requirements:

1. We have noticed that you have uploaded Supporting Information files, but you have not included a list of legends. Please add a full list of legends for your Supporting Information files after the references list. 

2. We note that your Data Availability Statement is currently as follows: "All data used in this analysis is publicly available and citations to the dataset are available through the reference list."

Additional Editor Comments (if provided):

Reviewers' comments:

Reviewer's Responses to Questions

**Comments to the Author**

1. Does this manuscript meet PLOS Global Public Health’s publication criteria ? Is the manuscript technically sound, and do the data support the conclusions? The manuscript must describe methodologically and ethically rigorous research with conclusions that are appropriately drawn based on the data presented.

Reviewer #1: Yes

Reviewer #2: Yes

2. Has the statistical analysis been performed appropriately and rigorously?

Reviewer #1: Yes

Reviewer #2: I don't know

3. Have the authors made all data underlying the findings in their manuscript fully available (please refer to the Data Availability Statement at the start of the manuscript PDF file)?

Reviewer #1: Yes

Reviewer #2: Yes

4. Is the manuscript presented in an intelligible fashion and written in standard English?

Reviewer #1: Yes

Reviewer #2: Yes

5. Review Comments to the Author

Reviewer #1: The manuscript “Countries’ Progress Towards Global Health Security Increased Health Systems Resilience During the Coronavirus Disease-19 Pandemic: A Difference-in-Difference Study of 191 Countries” leverages longitudinal data from around the world to explore the association between the Global Health Security Index (GHSI) and health systems resilience during times of crises—in this case the COVID-19 pandemic. Not only is this an incredibly important issue we need to know more about—the risk of public health emergencies continues and lessons from the last should be used for future policies—but their strategy of breaking down the index into its subcategories provides insight into areas ripe for future research.

The authors’ strategy of separating countries into treatment and control groups via one-step increments of their GHSI scores (overall and subcategories) and testing their association via a difference-in-difference model on childhood immunization coverage rates to show outcomes on health systems resilience is well thought out and posits a few very interesting concepts. One, breaking down complex indices is a viable approach to gain more specific understanding of a measure’s meaning. Two, health systems resilience’s positive association to environmental risks being more than double than the next closest subcategory (biosecurity) and having a much larger magnitude than the other components of the GHSI deserves further scrutiny by future researches and consideration by policymakers.

MAIN ISSUES

- Needs further clarification in the discussion section on how the results should be interpreted and how they align with current literature.

- Lack of possible mechanisms for the sub-component results undercuts their importance and is in detriment of the manuscript as a whole.

ISSUES

• Line 111: Data on childhood vaccination rates should be until 2022, if not you should clarify given that your results on the outcome are until 2022.

• Line 126: Clarify is data is missing or if the recommendation is not universal (Portugal only vaccinates BCG to children at risk, Austria it hasn’t been mandatory in a while, etc.)

• Lines 156-7: Consider briefly stating what the possible groups are (even if its explained later) since at this point it isn’t clear.

• Line 195: The table needs at a minimum clarification of the fact that both row and column wise percentages are being presented. Consider simplifying, while the n is important to know, the total column for the Income Group and Region are not that informative.

• Lines 260-1: Note provides insufficient information on what Control of Corruption means.

• Lines 286-9: Unclear, do you mean that less urban area reduce COVID deaths? Less deforestation and urbanization result in a higher environmental GHS and this is protective which is reflected by lower decline in vaccination? What are the potential mechanisms? Urbanization has more health care provider availability but a higher concentration of people during the COVID-19 pandemic led to an easier spread, so what is the mechanism here? Is this in line with current findings in the literature?

• Add if the study conforms to any relevant guidelines (CONSORT, MIAME, QUORUM, STROBE).

Reviewer #2: Thank you for the opportunity to review this interesting paper. The topic of the paper is critical as building health systems that are resilient to major shocks can prevent avoidable morbidity and mortality. In this study, the authors assess the relationship between preparedness and vaccine coverage as a proxy for resilience. They found that more prepared countries were able to improve vaccine coverage during the pandemic. While the paper is well-written, I think there should be several improvements to improve the clarity of the manuscript. The following are my comments:

Background

Lines 80 – 87: I think it is critical to define health system resilience in this section or the next. This is important point because there are many definitions of resilience in the literature and it would be important to describe resilience to the reader. It may also be helpful to briefly include what other studies have found to modulate resilience across countries and to briefly speculate why/how preparedness may also impact a countries’ resilience.

Lines 93: It would be helpful to briefly describe why childhood immunization coverage rates is an adequate proxy/measure of resilience.

Lines 75-78: The following a paper may also be a helpful citation of undercounting of COVID impacts impacting analyses: https://pubmed.ncbi.nlm.nih.gov/38879515/

Methods

Lines 110-111: For readers who may be unfamiliar, it would be helpful to know what age group the vaccine coverage estimates represent. For example, are the vaccine coverage estimates for children under 1 or under 15?

Line 112: In Figure S2, please add a caption/footnote describing what each blue line represents. I’m somewhat confused why there are two lines, and what average coverage rate means.

Line 152-155: Can you describe how the doubly-robust method was used here? I’m not an expert in DiD methods but from my understanding, you use a doubly-robust method to protect from model misspecification when you have a propensity logistic model and outcome model. But you don’t have a propensity score model here so why use a doubly-robust method? Please justify the use of this method and the benefit of the doubly-robust property in this instance?

Equation: Again, I’m inexperienced in the DiD method but it was my understanding that interaction terms between time and the intervention are needed in the outcome model. But there are no interaction terms in the equation, I assume this is needed to get the year-specific results in Table 3.

Lines 168-169: Why did the authorship team include the government indicators as a robustness check? Many of these indicators, especially the government effectiveness indicator, are already included in the GHS Index. So how is this additional analysis a robustness check? Please justify.

Results

Table 1: I believe this would be better communicated as a map showing what group each country belongs. Reading through the table may be too burdensome for readers.

The results section could be improved by describing trends in childhood vaccination rates and how rates were impacted during COVID. Perhaps included temporal trends of vaccinations may be helpful to improve the section.

Lines 201-202: It would be helpful to include an interpretation of your main effect size here rather than just providing the coefficient.

Many DiD studies generally present a time trend plot of the outcome by treatment group to show the effect of the intervention. I think this figure is needed to show readers the impact of preparedness on resilience.

Discussion

Please describe the mechanisms by which preparedness may improve resilience? Is that more prepared countries have pre-existing plans for maintaining essential health services during the pandemic? Perhaps they have a strong healthcare workforce and resources? Or perhaps they have fewer lockdowns? I think more discussion on why preparedness promotes vaccine services is needed to improve the impact of this paper.

More discussion on why the effect size is strongest in 2021 can further improve the discussion.

Minor: Can the authors briefly comment on the very high cutoff values for treatment effects in the analysis in the discussion? This implications for many countries because they would have substantial improvements to make

6. PLOS authors have the option to publish the peer review history of their article (what does this mean? ). If published, this will include your full peer review and any attached files.

**Do you want your identity to be public for this peer review?** For information about this choice, including consent withdrawal, please see our Privacy Policy .

Reviewer #1: No

Reviewer #2: No

---

## [Decision Letter · Decision Letter 1]

18 Nov 2024

PGPH-D-24-01417R1

Countries’ Progress Towards Global Health Security (GHS) Increased Health Systems Resilience During the Coronavirus Disease-19 (COVID-19) Pandemic: A Difference-in-Difference Study of 191 Countries

Dear Dr. Tozan,

Thank you for submitting your manuscript to PLOS Global Public Health. After careful consideration, we feel that it has merit but does not fully meet PLOS Global Public Health’s publication criteria as it currently stands. Therefore, we invite you to submit a revised version of the manuscript that addresses the points raised during the review process.

We look forward to receiving your revised manuscript.

Kind regards,

Sanjana Mukherjee, Ph.D., M.Sc.

Guest Editor

Journal Requirements:

Additional Editor Comments (if provided):

Reviewers' comments:

Reviewer's Responses to Questions

**Comments to the Author**

1. If the authors have adequately addressed your comments raised in a previous round of review and you feel that this manuscript is now acceptable for publication, you may indicate that here to bypass the “Comments to the Author” section, enter your conflict of interest statement in the “Confidential to Editor” section, and submit your "Accept" recommendation.

Reviewer #1: All comments have been addressed

Reviewer #2: All comments have been addressed

2. Does this manuscript meet PLOS Global Public Health’s publication criteria ? Is the manuscript technically sound, and do the data support the conclusions? The manuscript must describe methodologically and ethically rigorous research with conclusions that are appropriately drawn based on the data presented.

Reviewer #1: Yes

Reviewer #2: Yes

3. Has the statistical analysis been performed appropriately and rigorously?

Reviewer #1: Yes

Reviewer #2: Yes

4. Have the authors made all data underlying the findings in their manuscript fully available (please refer to the Data Availability Statement at the start of the manuscript PDF file)?

Reviewer #1: Yes

Reviewer #2: Yes

5. Is the manuscript presented in an intelligible fashion and written in standard English?

Reviewer #1: Yes

Reviewer #2: Yes

6. Review Comments to the Author

Reviewer #1: (No Response)

Reviewer #2: I thank the authors for their diligent edits to the manuscript. The authors responded to my comments with edits to the text and additional analyses. The paper elucidates on how health system resilience can be improved in the coming years to avert preventable morbidity and mortality from shocks. I just have the following small comments:

Lines 174-177: In reviewing the Anna and Zhao (2020) citation and the corresponding code in github, it looks like the authors did combine the propensity score weighting methods and outcome model adjustment during modeling. To better reflect this, I recommend re-writing this line with the following:

“The doubly-robust DiD estimator works by combining inverse probability of treatment weighting, estimated from a propensity score model, and outcome model adjustment to appropriately adjust for covariates and time-varying factors and generate a consistent estimate of the treatment effect. The inclusion of the same covariates in the propensity score model and outcome model protects from model misspecification.”

This language better communicates what was actually done in the study and the benefits of the DR DiD approach.

Line 244: Instead of using “significant” to describe the difference in coefficients, consider saying that the coefficient or effect size was larger compared to in 2020 or 2022.

7. PLOS authors have the option to publish the peer review history of their article (what does this mean? ). If published, this will include your full peer review and any attached files.

**Do you want your identity to be public for this peer review?** For information about this choice, including consent withdrawal, please see our Privacy Policy .

Reviewer #1: No

Reviewer #2: No

---

## [Editor Report · Decision Letter 2]

25 Nov 2024

Countries’ Progress Towards Global Health Security (GHS) Increased Health Systems Resilience During the Coronavirus Disease-19 (COVID-19) Pandemic: A Difference-in-Difference Study of 191 Countries

PGPH-D-24-01417R2

Dear Dr Tozan,

We are pleased to inform you that your manuscript 'Countries’ Progress Towards Global Health Security (GHS) Increased Health Systems Resilience During the Coronavirus Disease-19 (COVID-19) Pandemic: A Difference-in-Difference Study of 191 Countries' has been provisionally accepted for publication in PLOS Global Public Health.

Best regards,

Sanjana Mukherjee, Ph.D., M.Sc.

Guest Editor